# An update on *Scaponopselaphus* Scheerpeltz (Coleoptera: Staphylinidae) with the description of three new species, and a key to the species

Louri Klemann-Junior[1,2]*, Angélico Asenjo[3]*, Bruno Gouvea[4,5], Roberta M. Valente[4]

**1** Laboratório de Biodiversidade, Ecologia e Conservação de Ecossistemas Amazônicos, Universidade do Estado do Amazonas, Centro de Estudos Superiores de Itacoatiara, Itacoatiara, Amazonas, Brazil, **2** Programa de Pós-Graduação em Ciência e Tecnologia para Recursos Amazônicos, Instituto de Ciências Exatas e Tecnologia, Universidade Federal do Amazonas, Itacoatiara, Amazonas, Brazil, **3** Universidad Nacional Mayor de San Marcos, Departamento de Entomología del Museo de Historia Natural, Facultad de Ciencias Biológicas de la Universidad Nacional Mayor de San Marcos, Lima, Peru, **4** Laboratório de Invertebrados (LA-INV), Instituto de Ciências Biológicas, Universidade Federal do Pará, Belém, Pará, Brazil, **5** Programa de Pós-graduação em Zoologia (PPGZOOL), Instituto de Ciências Biológicas (ICB), Universidade Federal do Pará (UFPA), Belém, Pennsylvania, Brazil

\* klemannjr@yahoo.com.br (LKJ); angelico.asenjo@unmsm.edu.pe (AA)

## Abstract

Three new species of the genus *Scaponopselaphus* Scheerpeltz, 1972 are described; *Scaponopselaphus caribi* **sp. nov.**, and *Scaponopselaphus oby* **sp. nov.** from Brazil, and *Scaponopselaphus paradoxus* **sp. nov.** from Peru. New locality for *Scaponopselaphus mutator* (Sharp, 1876) is recorded. Major diagnostic features and pictorial and dichotomic identification key are provided for all species of the genus.

## Introduction

*Scaponopselaphus* Scheerpeltz is a genus of rove beetles distributed in the Amazon Rainforest (see Guayasamin et al. [1] for biome limits), with occurrence in Peru, Colombia, Guiana, Suriname, French Guiana, and Brazil at altitudes ranging from 30 to 317 m [2,3]. The genus, up to now, accommodates only two species and was revised by Chatzimanolis [2]. This genus is rarely collected and poorly represented in entomological collections. Chatzimanolis in his revision redescribed the type species of the genus, *Trigonopselaphus mutator* Sharp, 1876, described *Scaponopselaphus diaspartos* Chatzimanolis, 2015 from Colombia, and provided distribution maps and illustrations of morphological features.

Despite its wide distribution across several countries, *Scaponopselaphus* Scheerpeltz has scattered occurrences and is often separated by large gaps, such as the central region of the Amazon biome, suggesting under-sampling and/or occupation of a specific ecological niche. The limited presence of this taxon in collections highlights the challenge of understanding the true diversity of the genus.

**Data availability statement:** All relevant data are within the paper and its Supporting Information files.

**Funding:** The author(s) received no specific funding for this work.

**Competing interests:** The authors have declared that no competing interests exist.

In this article, we describe three new species of *Scaponopselaphus* Scheerpeltz, two from Brazil and one species from Peru, providing photographs, illustrations and measurements. We also provide a map of the known distribution of the genus, and a pictorial and dichotomic identification key to all species of the *Scaponopselaphus*.

## Materials and methods

### Morphological analyses

*Specimens*. All male specimen studied were relaxed in warm soapy water and the apical abdominal segments containing the aedeagus were dissected from the abdomen. First, the apical abdominal segments were cleared using 10% KOH. Then, the aedeagus was removed from the abdomen. The dissections and characters studies were carried out under a stereoscopic microscope. The photographs of *Scaponopselaphus caribi* **sp. nov.** were taken using a Canon 5D mark IV digital camera equipped with an extension tube and a MP-E 65 mm macro lens. The photographs of *Scaponopselaphus oby* **sp. nov.** and *Scaponopselaphus paradoxus* **sp. nov.** were taken using a stereo microscope Leica M205A equipped with a Leica DFC420 digital camera. Photographs of the aedeagus of *Scaponopselaphus caribi* **sp. nov.** were taken using a five-megapixel camera attached to a trinocular optical microscope. The images were automontaged on Helicon Focus 8 software. The adopted terminology for the descriptions follows Naomi [4,5] and Chatzimanolis [2].

### Characters and measurements

The description and measurements of the new species are based on the holotype, additional characters are included in the remarks section. In addition to the data previously described for the holotypes, we provided measurements for each structure (Fig 1) of all specimens (S1 Table), arranged by males and females in the measurements section for each species. The first number represents the maximum value, the second number represents the minimum value; when both values are the same, only a single value is shown. All measurements are in millimeters, expressed with two decimal places, and were taken under a Leica M205A (7.8X–128X) stereomicroscope (Fig 1).

In the type label data, quotation marks ("") separate different labels, and a vertical bar (|) separates different lines within a label. Text within square brackets [] is explanatory and it is not included in the original labels.

### Study area and occurrence records

The current study focuses on the South American continent, with 55 occurrence records compiled from: 19 records from Chatzimanolis [2]; 20 records from the Global Biodiversity Information Facility (GBIF) [6]; and 16 records from specimen labels studied in this publication. Records of *Scaponopselaphus* sp. refer to female specimens, whose species identification was not possible using only external morphological characters.

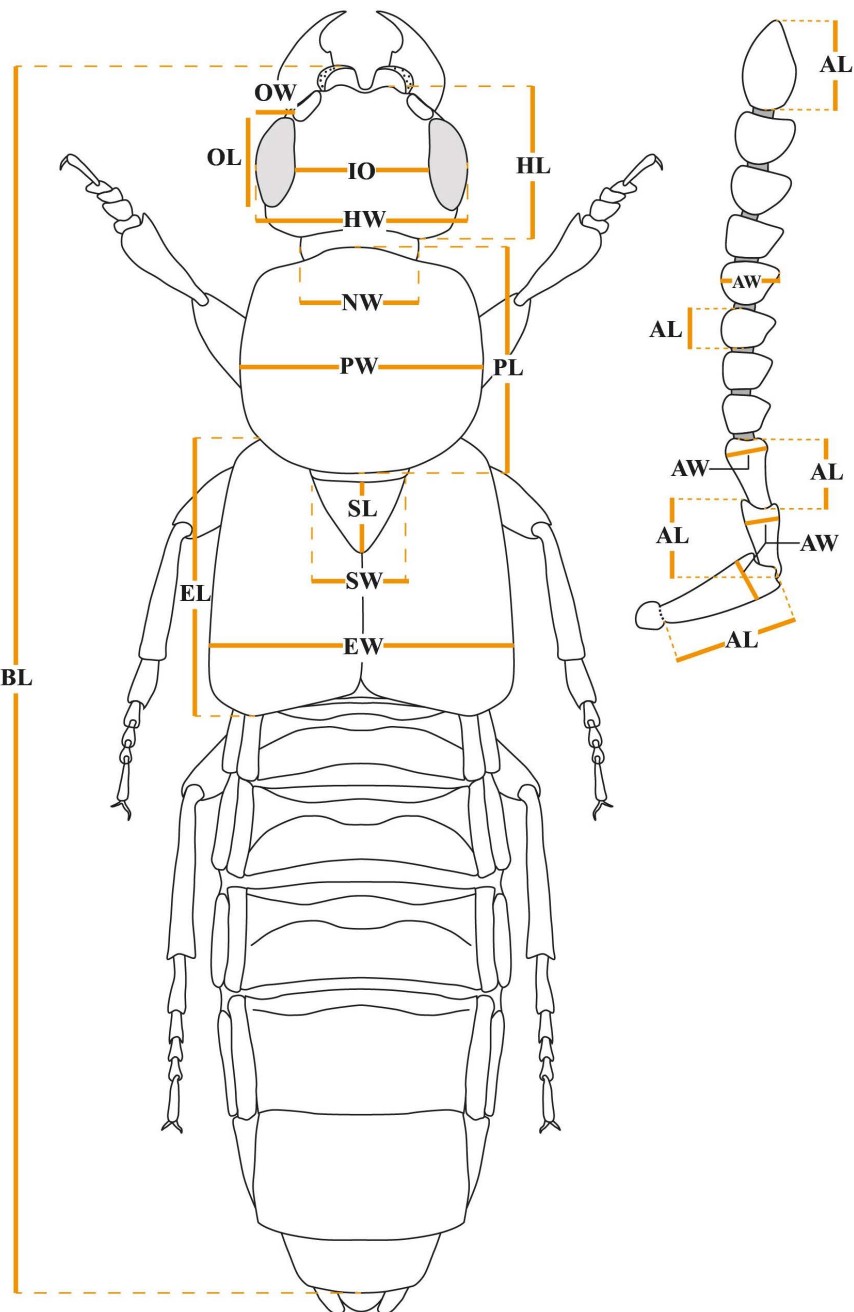

**Fig 1. *Scaponopselaphus* spp. measurements taken. Body: BL = body length, BW = maximum width of the elytra (=EW); Head: HL = maximum length of the head, HW = maximum width of the head including eyes, NW = maximum width of the neck, OL = maximum length of the eyes in dorsal view, OW = maximum width of the eyes in dorsal view, IO = minimum interocular distance; Antenna: AL = antennomere length; AW = antennomere width; Thorax: PL = pronotal length along the midline, PW = maximum width of the pronotum, EL = elytral length, EW = maximum width of the elytra, SL = scutellar shield length, SW = scutellar shield width.**

The current distribution map (Fig 2) was produced using QGIS 3.38 software [7] and a KMZ map [8] in S2 File.

**Depositaries**

The specimens studied are deposited in the following collections (curator(s) in parentheses).

**INPA:** Instituto Nacional de Pesquisas da Amazônia, Manaus, Amazonas, Brasil (Dr. Márcio Luiz de Oliveira).

**MPEG:** Museu Paraense Emilio Goeldi, Belém, Pará, Brazil (Dr. Orlando Tobias Silveira).

**MUSM:** Museo de Historia Natural, Universidad Nacional Mayor de San Marcos, Lima, Lima, Peru (Dr. Angélico Asenjo).

**UEA:** Entomological Collection of Centro de Estudos Superiores de Itacoatiara, Universidade do Estado do Amazonas, Itacoatiara, Amazonas, Brasil (Dr. Louri Kleman-Junior).

**UFPA:** Collection of Coleoptera of the Universidade Federal do Pará (Dra. Roberta Valente).

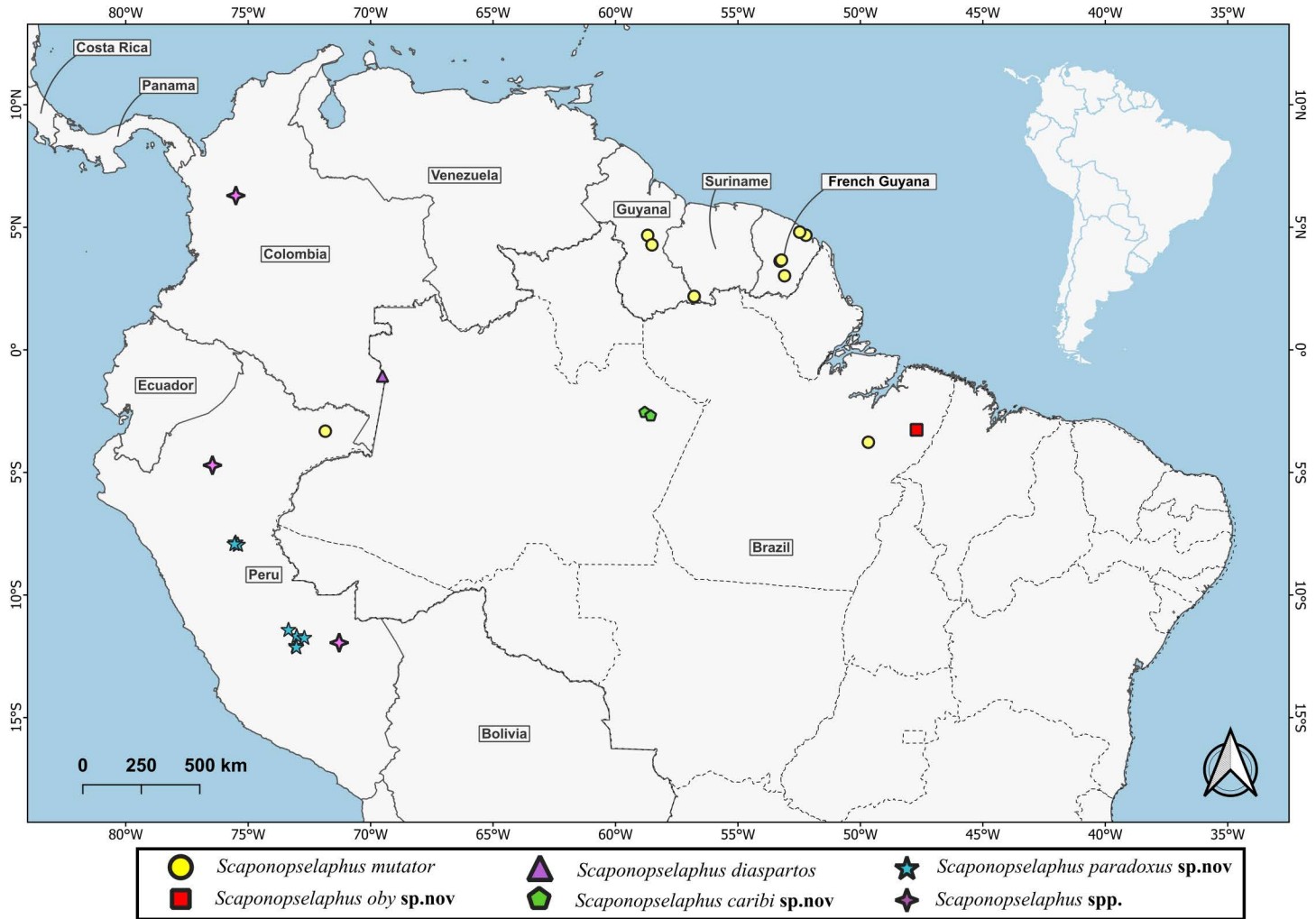

**Fig 2. Current distribution of *Scaponopselaphus* scheerpeltz.**

## Nomenclatural acts

The electronic edition of this article conforms to the requirements of the amended International Code of Zoological Nomenclature, and hence the new names contained herein are available under that Code from the electronic edition of this article. This published work and the nomenclatural acts it contains have been registered in ZooBank, the online registration system for the ICZN. The ZooBank LSIDs (Life Science Identifiers) can be resolved and the associated information viewed through any standard web browser by appending the LSID to the prefix "http://zoobank.org/". The LSID for this publication is: urn:lsid:zoobank.org:pub:C769B82C-66DD-4416-BBDC-CA37DCD944DD. The electronic edition of this work was published in a journal with an ISSN, and has been archived and is available from the following digital repositories: PubMed Central and LOCKSS.

## Results

Subfamily Staphylininae Latreille, 1802
Tribe Staphylinini Latreille, 1802
Subtribe Xanthopygina Sharp, 1884
Genus *Scaponopselaphus* Scheerpeltz, 1972
***Scaponopselaphus caribi* Klemann-Junior, Asenjo, Gouvea & Valente, sp. nov.** (LSID: urn:lsid:zoobank.org: act:60504E61-37B9-45BF-AA44-D0F2905263B5) (Figs 2,3A–C,4A–G,5A–F and 12).

 **Type material** (2 males, 1 female)

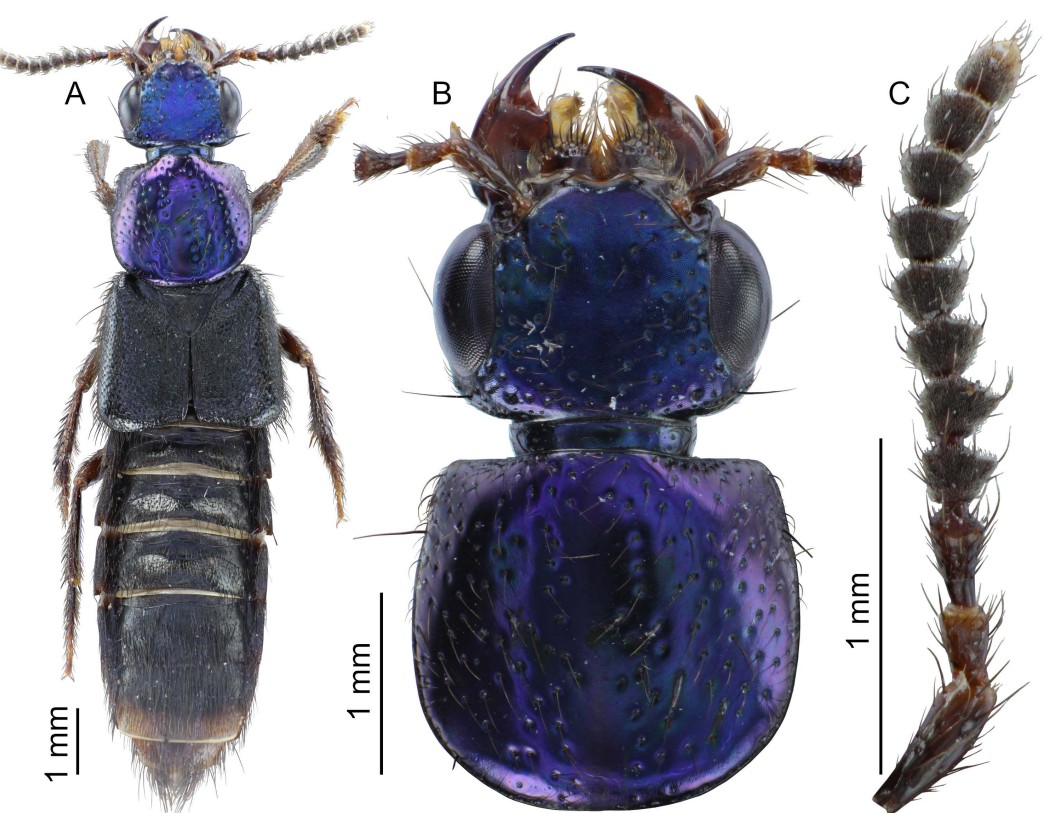

**Fig 3. Holotype of *Scaponopselaphus caribi* sp. nov. (INPA-COL 002848). (A) Dorsal habitus. (B) Head and pronotum. (C) Antenna.**

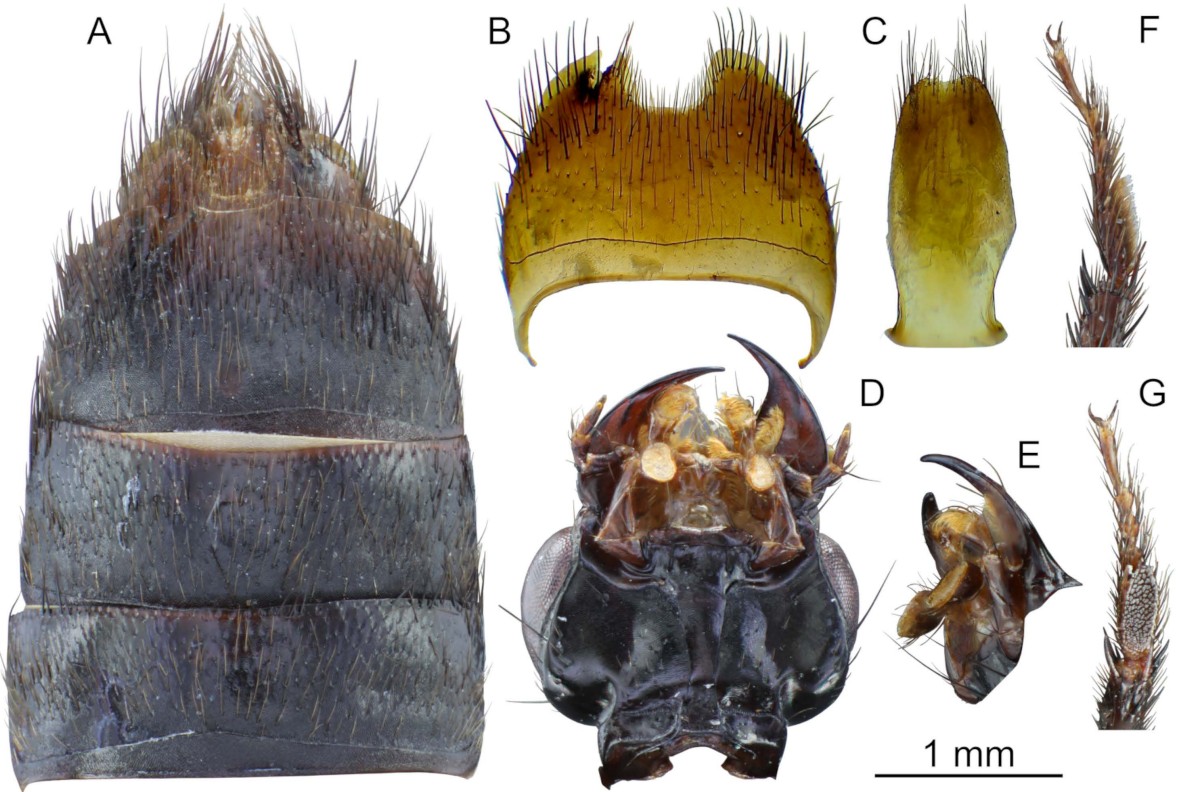

**Fig 4. Holotype of *Scaponopselaphus caribi* sp. nov. (INPA-COL 002848).** (A) Abdominal sternites V–IX. (B) Abdominal sternite VIII. (C) Abdominal sternite IX. (D) Head, ventral view. (E) Labial palpomeres, lateral view. (F) Mesotarsus, lateral view. (G) Mesotarsus, ventral view.

**Holotype.** male: "Brasil: AM[Amazonas state], Itapiranga[municipality] | Faz.[farm] Caribi - Prato rosa[pan trap - pink color] | 28.iv[April]-01.v[May].2022, T2/A1[trap code] | 2°32'50,48"S/58°48'23,16"W | L.Klemann-Junior | CESIT - UEA [left sideline]", "HOLOTYPE [red label] | *Scaponopselaphus* | *caribi* sp. nov. | Desig. Klemann-Junior *et al*. 2025", "INPA-COL 002848" (INPA).

**Paratypes.** 1 male, 1 female: "Brasil: AM [Amazonas state], Manaus, Embrapa[Brazilian Agricultural Research Corporation] | Guaraná convencional – mata[Conventional guarana cultivation – forest] | 11.x[October].2012 – Moerick trap | 2°53'42,18"S/59°59'10,58"W | K. Schoeninger" (male, INPA). "Brasil: AM [Amazonas state], Silves | Faz.[farm] Itapiranga | 04-11.v[May].2019 I[rainy season]11/12 [abbreviated logging year]T5A1 [trap code] | 2°41'25,31"S/58°33'42,82"W | L. F. B. Pereira | CESIT – UEA [left sideline]" (female, UEA). All paratypes with label: "PARATYPE [yellow label] | *Scaponopselaphus* | *caribi* sp. nov. | Desig. Klemann-Junior *et al*. 2025".

**Diagnosis.** Among the known *Scaponopselaphus* species, *Scaponopselaphus caribi* **sp. nov.** is the unique species that has a black elytra (Fig 3) and the apex of the lobes of paramere of aedeagus obliquely truncated (Fig 5).

**Measurements** males [min–max; number of specimens = 2]: ***Body***. BL = 10.88–11.29, BW = 2.72–2.76. ***Head***. HL = 1.33–1.35, HW = 1.80–1.87, NW = 1.01–1.05, OL = 0.74–0.85, OW = 0.31–0.35, IO = 1.18. ***Antenna***. antennomere 1 AL = 0.48–0.49, AW = 0.17, antennomere 2 AL = 0.24–0.25, AW = 0.15, antennomere 3 AL = 0.29–0.30, AW = 0.16, antennomere 4 AL = 0.17–0.20, AW = 0.20–0.22, antennomere 5 AL = 0.18–0.20, AW = 0.22–0.24, antennomere 6 AL = 0.17–0.18, AW = 0.22–0.24, antennomere 7 AL = 0.18, AW = 0.22–0.24, antennomere 8 AL = 0.17, AW = 0.21–0.23, antennomere 9

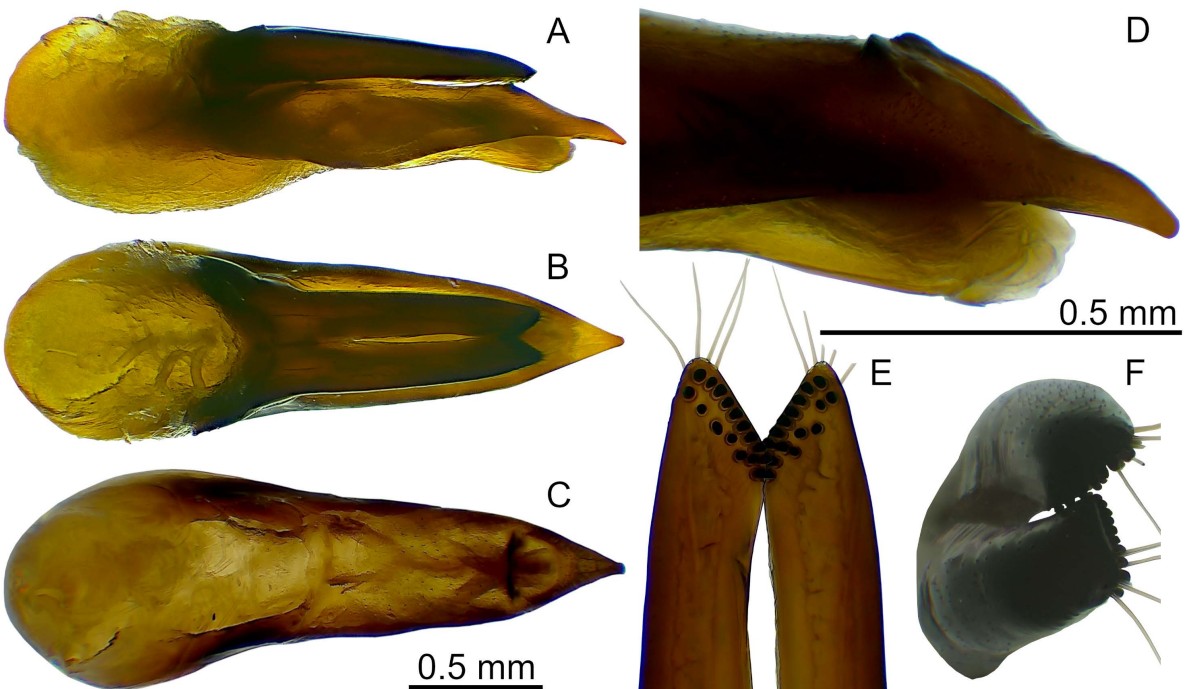

**Fig 5. Aedeagus of *Scaponopselaphus caribi* sp. nov. Holotype (INPA-COL 002848).** (A) Lateral view. (B) Parameral view. (C) Median lobe with paramere removed, parameral view. (D) Detail of the tip of median lobe, oblique view. (E) Detail of the tip of paramere, underside. (F) Detail of the tip of paramere, oblique view.

AL = 0.17, AW = 0.21–0.22, antennomere 10 AL = 0.18, AW = 0.20–0.21, antennomere 11 AL = 0.21–0.23, AW = 0.18. ***Thorax***. PL = 1.90–2.00, PW = 2.07–2.15, EL = 2.38–2.42, EW = 2.72–2.76, SL = 0.51–0.63, SW = 0.78–0.84.

**Measurements** female [number of specimens = 1]: ***Body***. BL = 11.13, BW = 2.67. ***Head***. HL = 1.23, HW = 1.78, NW = 1.03, OL = 0.75, OW = 0.33, IO = 1.13. ***Antenna***. antennomere 1 AL = 0.47, AW = 0.16, antennomere 2 AL = 0.26, AW = 0.14, antennomere 3 AL = 0.28, AW = 0.16, antennomere 4 AL = 0.16, AW = 0.21, antennomere 5 AL = 0.16, AW = 0.22, antennomere 6 AL = 0.16, AW = 0.22, antennomere 7 AL = 0.16, AW = 0.22, antennomere 8 AL = 0.17, AW = 0.23, antennomere 9 AL = 0.17, AW = 0.22, antennomere 10 AL = 0.18, AW = 0.20, antennomere 11 AL = 0.23, AW = 0.18. ***Thorax***. PL = 1.95, PW = 2.13, EL = 2.35, EW = 2.67, SL = 0.57, SW = 0.81.

**Description**. Holotype male, BL: 11.29, BW: 2.76.

**Coloration**. Head and pronotum dark metallic purple-blue (Fig 3A–3B). Antenna (Fig 3C) with antennomere 1 dark reddish brown with lighter apex, antennomere 2 reddish brown with lighter apex, antennomeres 3–10 dark reddish brown, and antennomere 11 dark reddish brown with yellowish apical third. Legs dark reddish brown (Fig 3A) with tarsi becomes lighter from tarsomere 1–4. Elytra (Fig 3A) black. Abdomen (Figs 3A and 4A) dark reddish brown, almost black, except posterior third of segment VII and segments VIII–IX reddish brown. Prosternum, meso- and metaventrite dark reddish brown.

**Head** (Fig 3A–3B) wider (HW: 1.87) than long (HL: 1.35) with rounded hind angles, clypeus emarginated, surface of vertex flat. Dark brown macrosetae along borders of head, with sparse umbilicate punctures each carrying dark brown microsetae, umbilicate punctures absent in middle and denser posteriorly. Integument of vertex with dense microsculpture formed by fingerprint-like microlines uniformly distributed. Anterior margin of frontoclypeal region emarginated (Fig 3B). Labial palpi with apical segment securiform with expanded apex (Fig 4D–4E). Eyes (Fig 3A–3B) large and prominent

(OL: 0.85; OW: 0.35), distance between eyes (IO) 1.18. Antennomeres (Fig 3C) 1–3 longer than wide, gradually club-like thickened, and without microtrichae; antennomere 1 (AL: 0.49; AW: 0.17), antennomere 2 (AL: 0.25; AW: 0.15), antennomere 3 (AL: 0.29; AW: 0.16); antennomeres 4–10 transverse and just slightly asymmetrical, antennomere 4 (AL: 0.17; AW: 0.22), antennomere 5 (AL: 0.18; AW: 0.24) antennomere 6 (AL: 0.17; AW: 0.24), antennomere 7 (AL: 0.18; AW: 0.24), antennomere 8 (AL: 0.17; AW: 0.23), antennomere 9 (AL: 0.17; AW: 0.22), antennomere 10 (AL: 0.18; AW: 0.21); antennomere 11 longer than wide (AL: 0.21; AW: 0.18); antennomeres 4–11 densely covered by microtrichae. Dorsal surface of neck with microsculpture like to vertex of head (Fig 3A–3B).

**Pronotum** (Fig 3A–B) wider than long (PL: 2.00; PW: 2.15), with anterior margin straight, lateral margin concave with hind lateral angles and posterior margin rounded. Disk with dark brown microsetae and, along borders, few dark brown macrosetae. Scattered medium size umbilicate punctures, denser near anterolateral corners; punctures absent along midline and near posterolateral corners. Integument shining, with dense microsculpture formed by fingerprint-like microlines uniformly distributed.

**Scutellum** (Fig 3A) with dense dark brown microsetae and punctures covering its surface, except posterolateral borders.

**Elytra** (Fig 3A) wider than pronotum (EL: 2.42; EW: 2.76), appearing shining; its dorsal surface with denser punctuation than pronotum with punctures almost confluent, and covered by dark brown setae and, along borders, by few dark brown macrosetae.

**Legs** (Fig 3A) with brown setae; protarsus dorsoventrally flattened and with spatulate pale setae ventrally; mesotarsomere 1 with spatulate pale setae ventrally (Fig 4F–4G). Mesotibia and metatibia with multiple rows of spurs on external side and with apical spurs. Protibia without multiple rows of spurs on external side; with single row of spurs apically.

**Abdomen** (Figs 3A and 4A) with dense and uniform punctuation pattern; punctures absent anteriorly to subbasal carina in segments III–V and anteriorly to basal carina in segment VI; each puncture with dark brown crosetae on segments III–VI and anteriorly to 2/3 of segment VII; yellowish crosetae on posterior third of segment VII and on segment VIII. Tergites and sternites with dense microsculpture formed by mesh-like microlines uniformly distributed. Few long dark brown macrosetae along lateral borders of all segments. Abdominal tergites III–VI with one undulating tergal basal carina; tergites III–V with subbasal one undulated carina; it is shallowest in tergite V. Sternite VII with round porose structure anteriorly (Fig 4A); its posterior margin with shallow and broad emargination at midline (Fig 4A); posterior margin of sternite VIII with deep U-shaped emargination at midline (Fig 4B); sternite IX with shallow U-shaped emargination at midline (Fig 4C). Tergite IX long and straight, and covered with long dark brown setae and with some dark macrosetae (Fig 4A).

**Aedeagus** as in Fig 5A–5F (length 2.32 mm). Paramere divided longitudinally in two lobes shorter than median lobe (Fig 5A–5B,5E–5F), and converging to obliquely truncated apex (Fig 5B and 5E), this last with peg setae arrangement in at most three rows (Fig 5E–5F). Sides of median lobe (Fig 5B–5C) converging to broad pointed apex, and with one small wide bicuspid tooth (Fig 5D), tapering apically in lateral view (Fig 5A and 5D).

**Female**. Similar to male but vertex of head slightly convex; head and pronotum lighter blue with green overtones, posterior margin of sternites VII and VIII straight, and mesotarsomere 1 without spatulate pale setae ventrally.

**Habitat**. Collected with Pan trap (pink color), with Moerick trap, and with Pitfall trap installed on the ground (baited with fresh human and pig feces 1:9 ratio) (see Moura et al. [9] for details). The capture sites belong to the company Precious Woods – Mil Madeiras Preciosas and are destined for selective logging. The vegetation in the area is Evergreen Tropical Forest "Floresta Ombrófila Densa de Terras Baixas" (Veloso et al. [10]).

**Known distribution** (Fig 2). Know from municipalities of Silves and Itapiranga in the state of Amazonas, Brazil.

**Etymology**. The specific epithet refers to the name of the river (Rio Caribi) near the collection site of the holotype. The name of the river is a reference to indigenous populations of Karib language family who inhabit the frontier region between Brazil (East Paru River, state of Pará), Surinam (Tapanahoni and Paloemeu rivers) and French Guiana (upper Maroni River and its tributaries the Tampok and Marouini). The specific epithet is a noun in apposition.

 

***Scaponopselaphus oby*** Asenjo, Gouvea, Valente & Klemann-Junior, **sp. nov.** (LSID: urn:lsid:zoobank.org: act:38675612-4D7F-4F5E-B9AE-85206EF56632) (Figs 2,6A–C,7A–H,8A–F and 12)

**Type material** (2 males, 1 female)

**Holotype.** male: "STP0358 | BRAZIL: Paragominas[Municipality], | Mineradora Hydro. AL9, | S 03°15'43" W 047°42'07". | 8-V[May]-18[2018]. p. amarelo[Pan Trap - yellow color]" "STP0360 | BRAZIL: Paragominas, | Mineradora Hydro. AL2, | S 03°15'44" W 047°42'26". | 05-V[May]-18[2018]. Winkler", "HOLOTYPE [red label] | *Scaponopselaphus* | *oby* sp. nov. | Desig. Klemann-Junior *et al*. 2025" (MPEG).

**Paratypes.** 1 male, 1 female: "STP0360 | BRAZIL: Paragominas[Municipality], | Mineradora Hydro. AL2, | S 03°15'44" W 047°42'26". | 05-V[May]-18[2018]. Winkler" (1 male, UFPA).

"STP0357 | BRAZIL: Paragominas[Municipality], | Mineradora Hydro. AL9, | S 03°15'43" W 047°42'07". | 8-V[May]-18[2018]. p. amarelo[Pan Trap - yellow color]" (1 female, UFPA).

All paratypes with label: "PARATYPE [yellow label] | *Scaponopselaphus* | *oby* sp. nov. | Desig. Klemann-Junior *et al*. 2025".

**Diagnosis.** *Scaponopselaphus oby* **sp. nov.** is similar to *Scaponopselaphus diaspartos* Chatzimanolis by the coloration brown of elytra (Fig 6A) and the peg setae on the lobes of paramere of aedeagus scattered throughout its length (Fig 8E–8F). However, *Scaponopselaphus oby* **sp. nov.** can be differentiated from *Scaponopselaphus diaspartos* Chatzimanolis by the apex of the lobes of paramere of aedeagus not reaching the apex of median lobe as Fig 8A–8B (the apex of paremere of aedeagus reaching the apex of median lobe in *Scaponopselaphus diaspartos* Chatzimanolis (Fig 6A–6B in Chatzimanolis [2])). *Scaponopselaphus oby* **sp. nov.** can also be distinguished from the remaining species in the genus by the presence of a triangular tooth on the middle portion of the median lobe of the aedeagus (Fig 8D).

**Measurements** males [min–max (average); number of specimens = 2]: ***Body***. BL = 11.55, BW = 2.24. ***Head***. HL = 1.27–1.30, HW = 1.87–1.88, NW = 0.99–1.00, OL = 0.82–0.83, OW = 0.33–0.34, IO = 1.16–1.18. ***Antenna***. antennomere 1 AL = 0.51, AW = 0.18, antennomere 2 AL = 0.29, AW = 0.15–0.16, antennomere 3 AL = 0.29–0.30, AW = 0.17–0.18, antennomere 4 AL = 0.16–0.17, AW = 0.21–0.22, antennomere 5 AL = 0.15, AW = 0.22, antennomere 6 AL = 0.16, AW = 0.22–0.23, antennomere 7 AL = 0.15–0.16, AW = 0.22, antennomere 8 AL = 0.15–0.17, AW = 0.22, antennomere 9 AL = 0.16, AW = 0.21–0.22, antennomere 10 AL = 0.16–0.17, AW = 0.21, antennomere 11 AL = 0.27–0.28, AW = 0.19. ***Thorax***. PL = 1.87–2.05, PW = 1.87–2.05, EL = 2.22, EW = 2.24, SL = 0.65–0.83, SW = 0.79.

**Measurements** female [number of specimens = 1]: ***Body***. BL = 10.95, BW = 2.45. ***Head***. HL = 1.21, HW = 1.75, NW = 0.99, OL = 0.79, OW = 0.32, IO = 1.11. ***Antenna***. antennomere 1 AL = 0.40, AW = 0.19, antennomere 2 AL = 0.27, AW = 0.15, antennomere 3 AL = 0.29, AW = 0.15, antennomere 4 AL = 0.13, AW = 0.20, antennomere 5 AL = 0.15, AW = 0.21, antennomere 6 AL = 0.13, AW = 0.21, antennomere 7 AL = 0.14, AW = 0.21, antennomere 8 AL = 0.14, AW = 0.20, antennomere 9 AL = 0.14, AW = 0.21, antennomere 10 AL = 0.14, AW = 0.21, antennomere 11 AL = 0.24, AW = 0.19. ***Thorax***. PL = 1.80, PW = 1.90, EL = 2.24, EW = 2.45, SL = 0.61, SW = 0.74.

**Description.** Holotype male, BL: 11.55, BW: 2.24.

**Coloration.** Dorsal head and pronotum metallic green (Fig 6A–6B). Legs including tarsi dark brown. Elytra (Fig 6A) dark brown. Abdomen (Fig 6A) dark brown, except apex of segment VII, and segments VIII–IX reddish brown. Prosternum, meso- and metaventrite from dark reddish brown to black.

**Head** (Fig 6A–6B) wider (HW: 1.88) than long (HL: 1.27) with long dark brown macrosetae along borders; dorsal surface with sparse umbilicate punctures each carrying dark brown setae, umbilicate punctures absent anteriorly in middle and denser posteriorly. Rounded hind angles, clypeus strongly emarginated, surface of vertex flat. Integument of vertex with dense microsculpture formed by fingerprint-like microlines uniformly distributed. Anterior margin of frontoclypeal region slightly emarginated (Fig 6B). Labial palpi with apical segment securiform with expanded apex (Fig 7D–7F). Eyes (Fig 6A–6B) large and prominent (OL: 0.83; OW: 0.34), distance between eyes (IO) 1.16. Antenna (Fig 6C) with antennomeres 1–3 longer than wide, gradually club-like thickened, and without microtrichae; antennomere 1 (AL: 0.51;

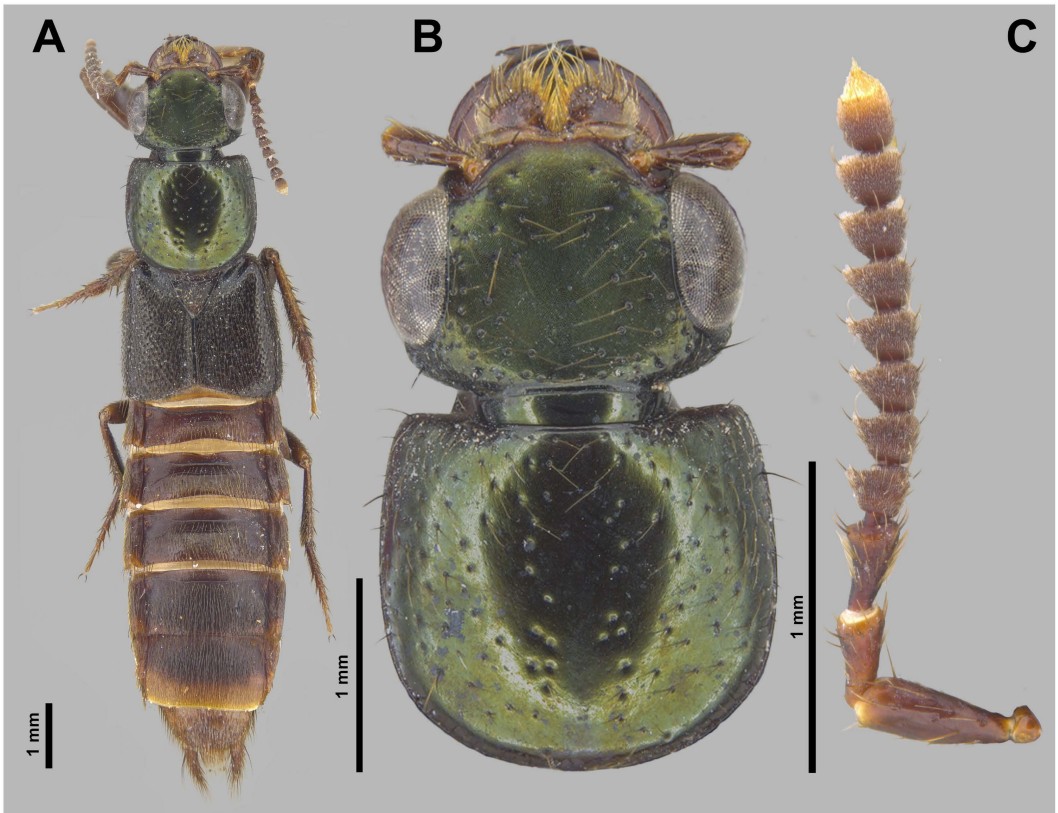

**Fig 6. Holotype of *Scaponopselaphus oby* sp. nov. (STP0358).** (A) Dorsal habitus. (B) Head and pronotum. (C) Antenna.

AW: 0.18), antennomere 2 (AL: 0.29; AW: 0.15), antennomere 3 (AL: 0.29; AW: 0.18); antennomeres 4–10 transverse and just slightly asymmetrical, antennomere 4 (AL: 0.16; AW: 0.21), antennomere 5 (AL: 0.15; AW: 0.22), antennomere 6 (AL: 0.16; AW: 0.22), antennomere 7 (AL: 0.15; AW: 0.22), antennomere 8 (AL: 0.15; AW: 0.22), antennomere 9 (AL: 0.16; AW: 0.21), antennomere 10 (AL: 0.16; AW: 0.21); antennomere 11 longer than wide (AL: 0.28; AW: 0.19); antennomeres 4–11 densely covered by microtrichae. Dorsal surface of neck with microsculpture like vertex of head and with few micropunctures (Fig 6B).

   **Pronotum** (Fig 6A–6B) wider than long (PL: 1.87; PW: 2.05) with straight anterior and lateral margins, and hind angles rounded. Disk with rounded umbilicate punctures, except on medial line; umbilicate punctures with dark brown setae. Integument shining, with dense microsculpture in form of rooves between umbilicate punctures.

   **Scutellum** (Fig 6A) with dense dark brown setae and micro-punctures covering surface, except its posterolateral borders.

   **Elytra** (Fig 6A) wider than pronotum (EL: 2.22; EW: 2.24); surface polished and shiny, with denser punctuation than pronotum with punctures almost confluent; and covered by dark brown setae and, along borders with few dark brown macrosetae.

   **Legs** (Fig 6A) with brown setae; protarsus dorsoventrally flattened and with spatulate pale setae ventrally; mesotarsomere 1 with spatulate pale setae ventrally (Fig 7G–7H). Mesotibia and metatibia with multiple rows of spurs on external side, and with apical spurs. Protibia without multiple rows of spurs on external side, and with single row of spurs apically.

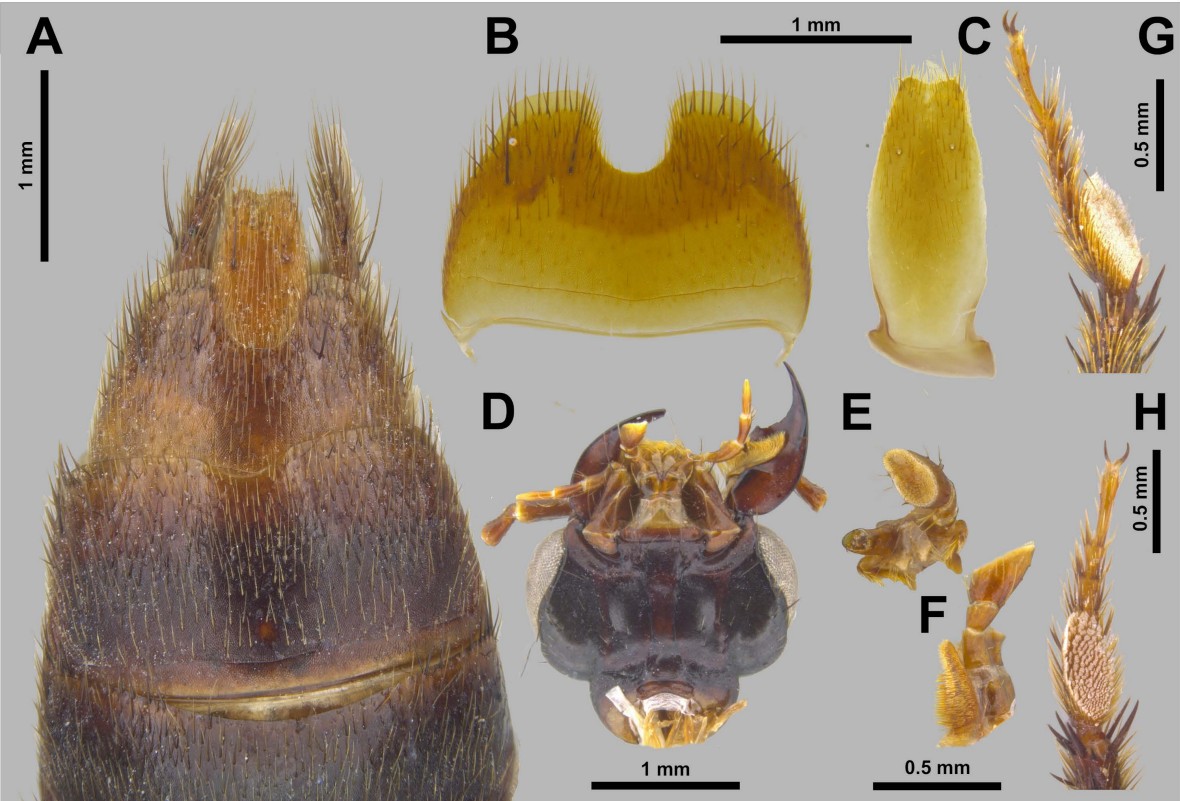

**Fig 7. Holotype of *Scaponopselaphus oby* sp. nov. (STP0358).** (A) Abdominal sternites VI–IX. (B) Abdominal sternite VIII. (C) Abdominal sternite IX. (D) Head, ventral view. (E) Labial palpomeres, anterior view. (F) Labial palpomeres, lateral view. (G) Mesotarsus, lateral view. (H) Mesotarsus, ventral view.

**Abdomen** (Figs 6A and 7A) abdominal tergites III–VI with tergal basal carina; tergites III–V with subbasal curved carina. Sternite VII with round porose structure anteriorly (Fig 7A); its posterior margin with shallow and broad emargination at midline (Fig 7A); posterior margin of sternite VIII with deep U-shaped emargination at midline (Fig 7A–7B); sternite IX with shallow U-shaped emargination at midline (Fig 7C). Tergite IX long and straight, and covered with long dark brown setae and some dark macrosetae (Fig 7A).

**Aedeagus** as in Fig 8A–8F (length 2.24 mm). Paramere (Fig 8B) divided longitudinally in two lobes slightly shorter than median lobe (Fig 8A–8B), and converging to pointed apex (Fig 8B–8E) slightly convex, and with peg setae situated at middle until half of length (Fig 8E–8F). Sides of median lobe converging to narrow pointed apex (Fig 8B–8C); this last with shallow median longitudinal carina and one small tooth on each side (Fig 8D), middle portion of median lobe with one triangular tooth (Fig 8D).

**Female**. Similar to male, but vertex of head slightly convex, posterior margin of sternites VII and VIII straight, and mesotarsomere 1 without spatulate pale setae ventrally.

**Habitat**. The specimens were collected with yellow trap in Amazon Rainforest near the area of mining activity of bauxite extraction. For a complete description of the area to see Gouvea et al. [11].

**Known distribution** (Fig 2). Know from municipalities of Paragominas in the state of Pará, Brazil.

**Etymology**. The epithet name "oby" is a noun in apposition taken from the native language Tupi-Guarani and means "green or blue", due to the color of the head and pronotum.

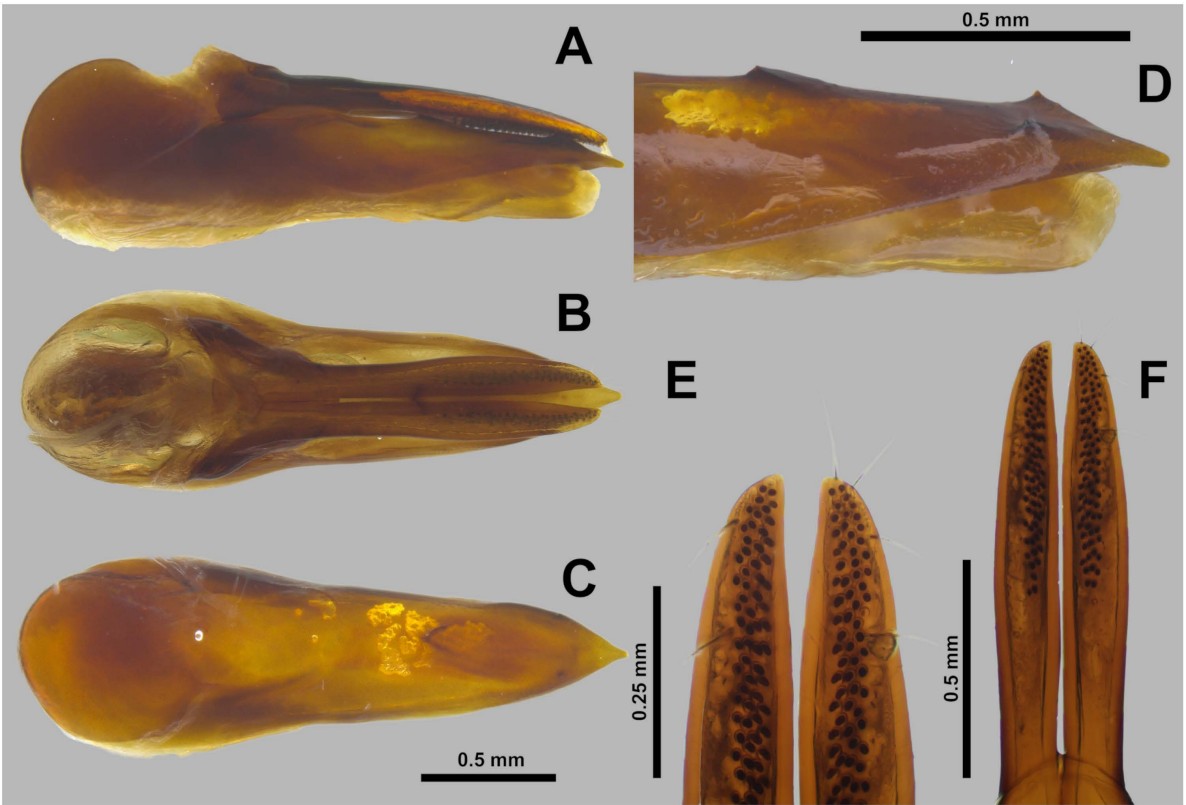

**Fig 8. Aedeagus of *Scaponopselaphus oby* sp. nov., Holotype (STP0358).** (A) Lateral view. (B) Parameral view. (C) Median lobe with paramere removed, parameral view. (D) Detail of the tip of median lobe, oblique view. (E) Detail of the tip of paramere, underside. (F) Paramere, underside.

**Remarks**. Between the male and female specimens studied, there is a variation in the metallic overtones, ranging from green to blue in the head and pronotum.

*Scaponopselaphus paradoxus* **Asenjo, Gouvea, Valente & Klemann-Junior, sp. nov.** (LSID: urn:lsid:zoobank.org: act:B2B21752-0435-456E-89BC-A8E47795DA0F) (Figs 2,9A–C,10A–G,11A–F and 12)

**Type material** (6 males, 7 females)

**Holotype.** male: "PERU: LO[Department of Loreto], Ucayali, Río Pisqui, | 7°55'44.85"S/ 75°33'34.66"O, | 202 m, 6-8.x[October].2011, L. Sulca", "[image Data Matrix barcode]MUSM-ENT | 0150355" (male MUSM). "HOLOTYPE [red label] | *Scaponopselaphus* | *paradoxus* sp. nov. | Desig. Klemann-Junior *et al.* 2025" (MUSM).

**Paratypes.** 5 males, 7 females:

"PERU: LO[Department of Loreto], Ucayali, | Contamana, Río Pisqui, | 7°57'22.94"S, | 75°25'8.28"W, 216 m, 1- | 3.x[October].2011, L. Sulca leg.", "[image Data Matrix barcode]MUSM-ENT | 0150349" (male, MUSM).

"PERU: LO[Department of Loreto], Ucayali, | Contamana, Río Pisqui, | 7°57'22.94"S, | 75°25'8.28"W, 216 m, 1- | 3.x[October].2011, L. Sulca leg.", "[image Data Matrix barcode]MUSM-ENT | 0150350" (male, MUSM).

"PERU: LO[Department of Loreto], Ucayali, | Contamana, Río Pisqui, | 7°57'22.94"S, | 75°25'8.28"W, 216 m, 1- | 3.x[October].2011, L. Sulca leg.", "[image Data Matrix barcode]MUSM-ENT | 0150351" (female, MUSM).

"PERU: LO[Department of Loreto], Ucayali, | Contamana, comunidad | nativa Roaboya, | 7°51'54.25"S, | 75°30'53.55"W, 222 m, 3- | 5.x[October].2011, L. Sulca leg." "[image Data Matrix barcode]MUSM-ENT | 0150352" (female, MUSM).

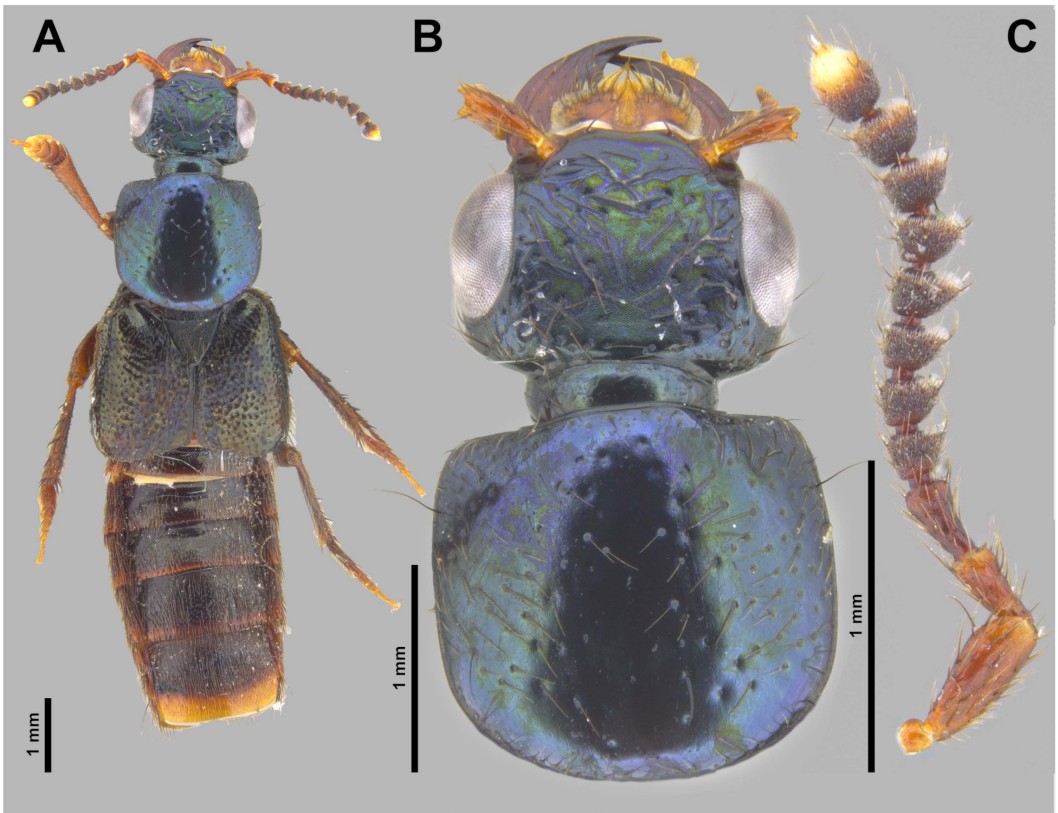

**Fig 9. Holotype of *Scaponopselaphus paradoxus* sp. nov. (MUSM-ENT 0150355).** (A) Dorsal habitus. (B) Head and pronotum. (C) Antenna.

"PERU: LO[Department of Loreto], Ucayali, | Contamana, Río Pisqui, | 7°57'22.94"S, | 75°25'8.28"W, 216 m, 1- | 3.x[October].2011, L. Sulca leg.", "[image Data Matrix barcode]MUSM-ENT | 0150353" (female, MUSM).

"PERU: LO[Department of Loreto], Ucayali, | Contamana, Río Pisqui, | 7°57'22.94"S, | 75°25'8.28"W, 216 m, 1- | 3.x[October].2011, L. Sulca leg.", "[image Data Matrix barcode]MUSM-ENT | 0150354" (female, MUSM).

"PERU: LO[Department of Loreto], Ucayali, Río Pisqui, | 7°55'44.85"S/ 75°33'34.66"O, | 202 m, 6-8.x[October].2011, L. Sulca", "[image Data Matrix barcode]MUSM-ENT | 0150356" (female, MUSM).

"PERÚ: CU[Department of Cuzco], La | Convención, Megantoni, | 11°41'16"S/ 73°01'12"O | 13.iii[March].2022 346m, S. Bejar", "[image Data Matrix barcode]MUSM-ENT | 0150346" (male, MUSM).

"PERÚ: CU[Department of Cuzco], La Convención, | Echarate, San Martín Norte. | 11°45'18.8"S/72°42'26"W, | 430m.10-14.xi[November].2010. B. | Medina y Z. Bravo.", "[image Data Matrix barcode]MUSM-ENT | 0150347" (female, MUSM).

"PERÚ: Cu[Department of Cuzco], La Convención, | Echarate, 558 m Sagari | 11°25'46.62"S/73°21'16.47"O, | 23.xi[November].2018, E.Gamboa", "[image Data Matrix barcode]MUSM-ENT | 0150348" (female, MUSM).

"PERU: CU[Department of Cuzco], La Convención, | Reserva Comunal | Matsiguenga, 581m, | 12°08'32.8"S,73°02'03.5"W, | 02.ii[February].2007, C.Castillo | Guadua-mixed forest | Pitfall trap | PM03.070302.P09", "[image Data Matrix barcode]MUSM-ENT | 0150357" (male, MUSM).

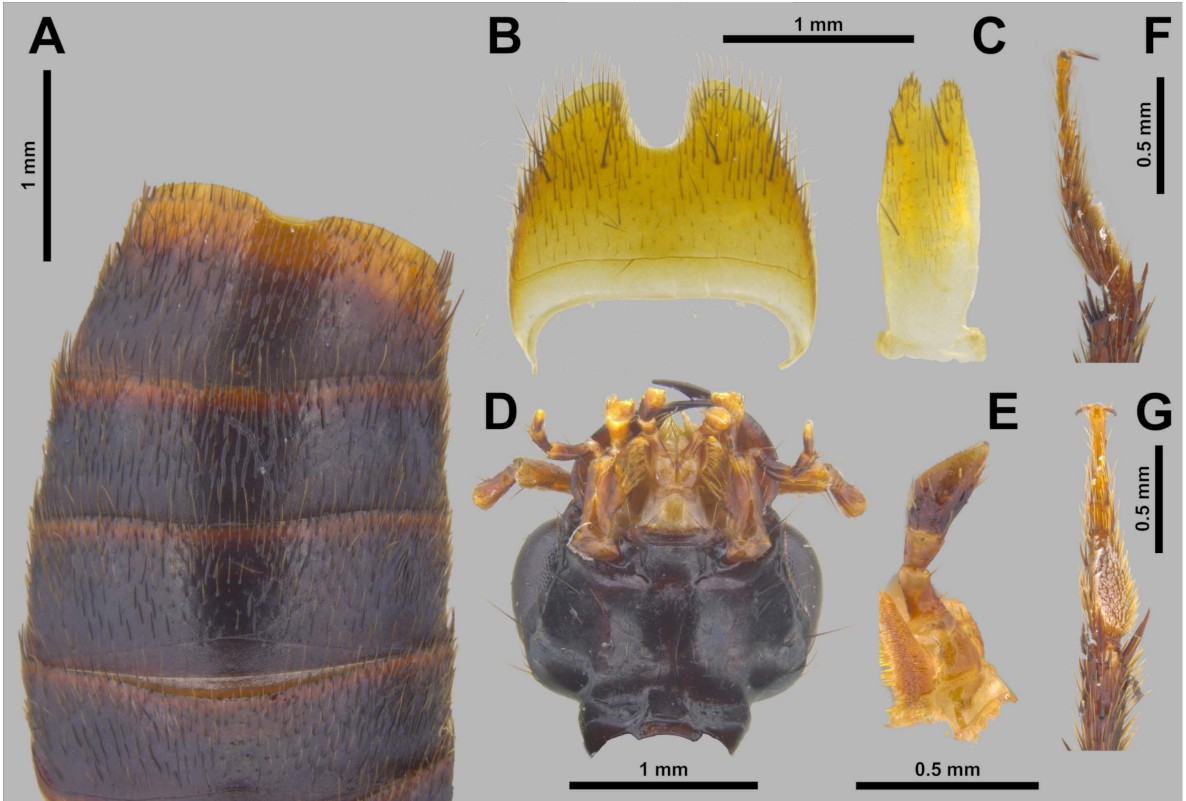

**Fig 10. Holotype of *Scaponopselaphus paradoxus* sp. nov. (MUSM-ENT 0150355).** (A) Abdominal sternites V–VII. (B) Abdominal sternite VIII. (C) Abdominal sternite IX. (D) Head, ventral view. (E) Labial palpomeres, lateral view. (F) Mesotarsus, lateral view. (G) Mesotarsus, ventral view.

"PERU: CU[Department of Cuzco], La Convención | 12°05'7.75"S 73°03'08.06"W | 640m, 22.iv[April].2007, F. Azorsa | Bosques de colinas bajas | Yellow pan trap | PM06.070422.Y08", "[image Data Matrix barcode]MUSM-ENT | 0150358" (male, MUSM).

All paratypes with label: "PARATYPE [yellow label] | *Scaponopselaphus* | *paradoxus* sp. nov. | Desig. Klemann-Junior *et al*. 2025".

**Diagnosis.** *Scaponopselaphus paradoxus* **sp. nov.** is similar to *Scaponopselaphus mutator* (Sharp) due to the brown color of the elytra (Fig 9A) and the peg setae concentrated near the apex of the lobes of paramere of aedeagus (Fig 10E). However, *Scaponopselaphus paradoxus* **sp. nov.** can be differentiated from *Scaponopselaphus mutator* (Sharp) by the pointed apex of the lobes of paramere of aedeagus (Fig 10E–10F) (rounded apex of the lobes of paramere in *Scaponopselaphus mutator* (Sharp) (Fig 7B–7C in Chatzimanolis [2])).

**Measurements** males [min–max; number of specimens = 6]: ***Body***. BL = 10.14–11.81, BW = 2.25–2.72. ***Head***. HL = 1.07–1.25, HW = 1.53–1.75, NW = 0.85–0.96, OL = 0.68–0.82, OW = 0.26–0.33, IO = 0.99–1.10. ***Antenna***. antennomere 1 AL = 0.39–0.50, AW = 0.16–0.20, antennomere 2 AL = 0.22–0.29, AW = 0.12–0.15, antennomere 3 AL = 0.22–0.27, AW = 0.15–0.16, antennomere 4 AL = 0.14–0.15, AW = 0.15–0.19, antennomere 5 AL = 0.13–0.16, AW = 0.18–0.22, antennomere 6 AL = 0.12–0.15, AW = 0.18–0.22, antennomere 7 AL = 0.12–0.16, AW = 0.18–0.23, antennomere 8 AL = 0.13–0.15, AW = 0.19–0.23, antennomere 9 AL = 0.11–0.16, AW = 0.19–0.23, antennomere 10 AL = 0.13–0.17, AW = 0.20–0.24, antennomere 11 AL = 0.26–0.30, AW = 0.18–0.21. ***Thorax***. PL = 1.58–1.91, PW = 1.70–1.94, EL = 1.91–2.41, EW = 2.25–2.72, SL = 0.52–0.67, SW = 0.63–0.76.

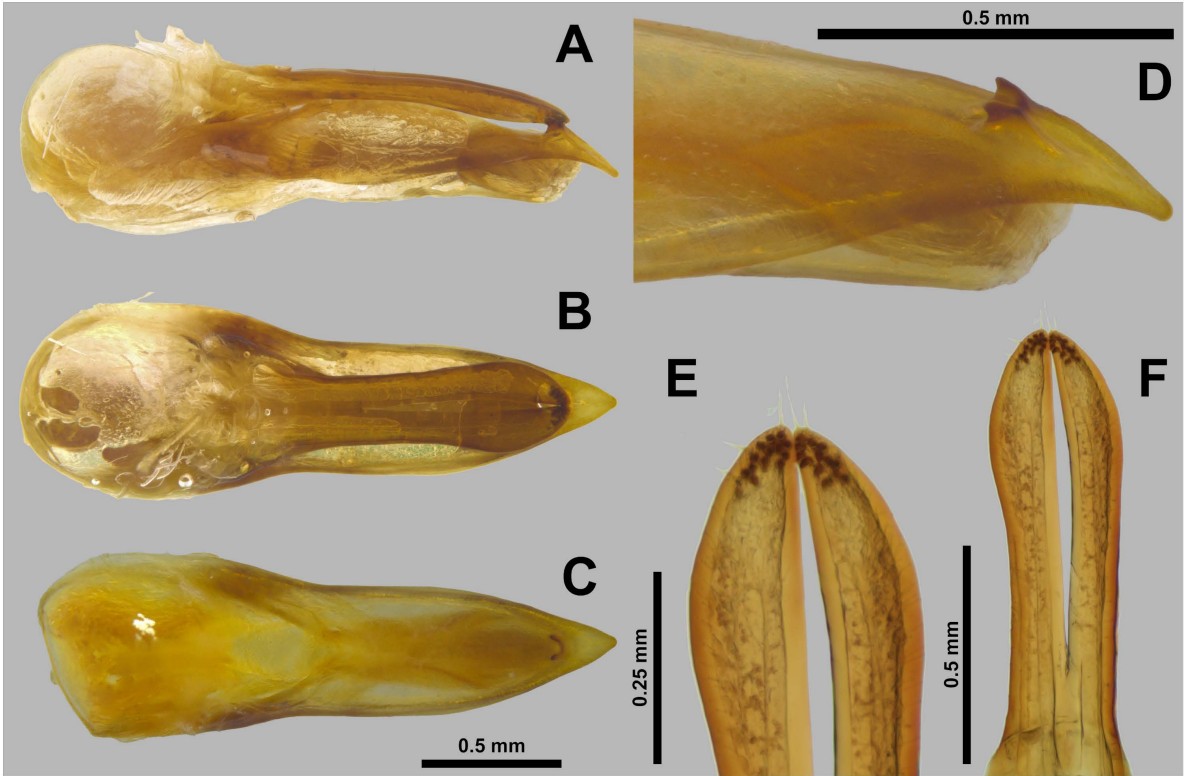

**Fig 11. Aedeagus of *Scaponopselaphus paradoxus* sp. nov. Holotype (MUSM-ENT 0150355).** (A) Lateral view. (B) Parameral view. (C) Median lobe with paramere removed, parameral view. (D) Detail of the tip of median lobe, oblique view. (E) Detail of the tip of paramere, underside. (F) Paramere, underside.

**Measurements** females [min–max; number of specimens = 7]: ***Body***. BL = 9.84–11.48, BW = 2.46–2.76. ***Head***. HL = 1.17–1.25, HW = 1.65–1.77, NW = 0.89–0.98, OL = 0.72–0.84, OW = 0.26–0.33, IO = 1.05–1.11. ***Antenna***. antennomere 1 AL = 0.45–0.50, AW = 0.16–0.19, antennomere 2 AL = 0.22–0.30, AW = 0.13–0.15, antennomere 3 AL = 0.19–0.27, AW = 0.15–0.16, antennomere 4 AL = 0.13–0.15, AW = 0.17–0.19, antennomere 5 AL = 0.13–0.14, AW = 0.18–0.21, antennomere 6 AL = 0.12–0.15, AW = 0.20–0.22, antennomere 7 AL = 0.12–0.15, AW = 0.20–0.21, antennomere 8 AL = 0.13–0.14, AW = 0.21–0.22, antennomere 9 AL = 0.12–0.15, AW = 0.20–0.23, antennomere 10 AL = 0.13–0.16, AW = 0.20–0.24, antennomere 11 AL = 0.24–0.27, AW = 0.18–0.21. ***Thorax***. PL = 1.72–1.91, PW = 1.92–2.00, EL = 2.12–2.42, EW = 2.46–2.76, SL = 0.61–0.71, SW = 0.57–0.78.

**Description.** Holotype male, BL: 10.27, BW: 2.52.

**Coloration.** Dorsal head and pronotum metallic blue with overtones green (Fig 9A–9B). Legs including tarsi light brown. Elytra (Fig 9A) dark brown. Abdomen (Fig 9A) dark brown except apex of segment VII, and segments VIII–IX light brown. Prosternum, meso- and metaventrite from dark reddish brown to black.

**Head** (Fig 9B) wider (HW: 1.71) than long (HL: 1.16), with rounded hind angles, its dorsal surface with long dark brown macrosetae along borders, and with sparse umbilicate punctures each carrying dark brown setae, umbilicate punctures absent anteriorly in middle and denser posteriorly. Integument of vertex with dense microsculpture formed by fingerprint-like microlines uniformly distributed. Clypeus strongly emarginated until its base, surface of vertex flat. Anterior margin of frontoclypeal region straight (Fig 9B). Labial palpi with apical segment securiform with expanded apex (Fig 10D–10E). Eyes (Fig 9A–9B) large and prominent (OL: 0.78; OW: 0.32), distance between eyes (IO) 1.09.

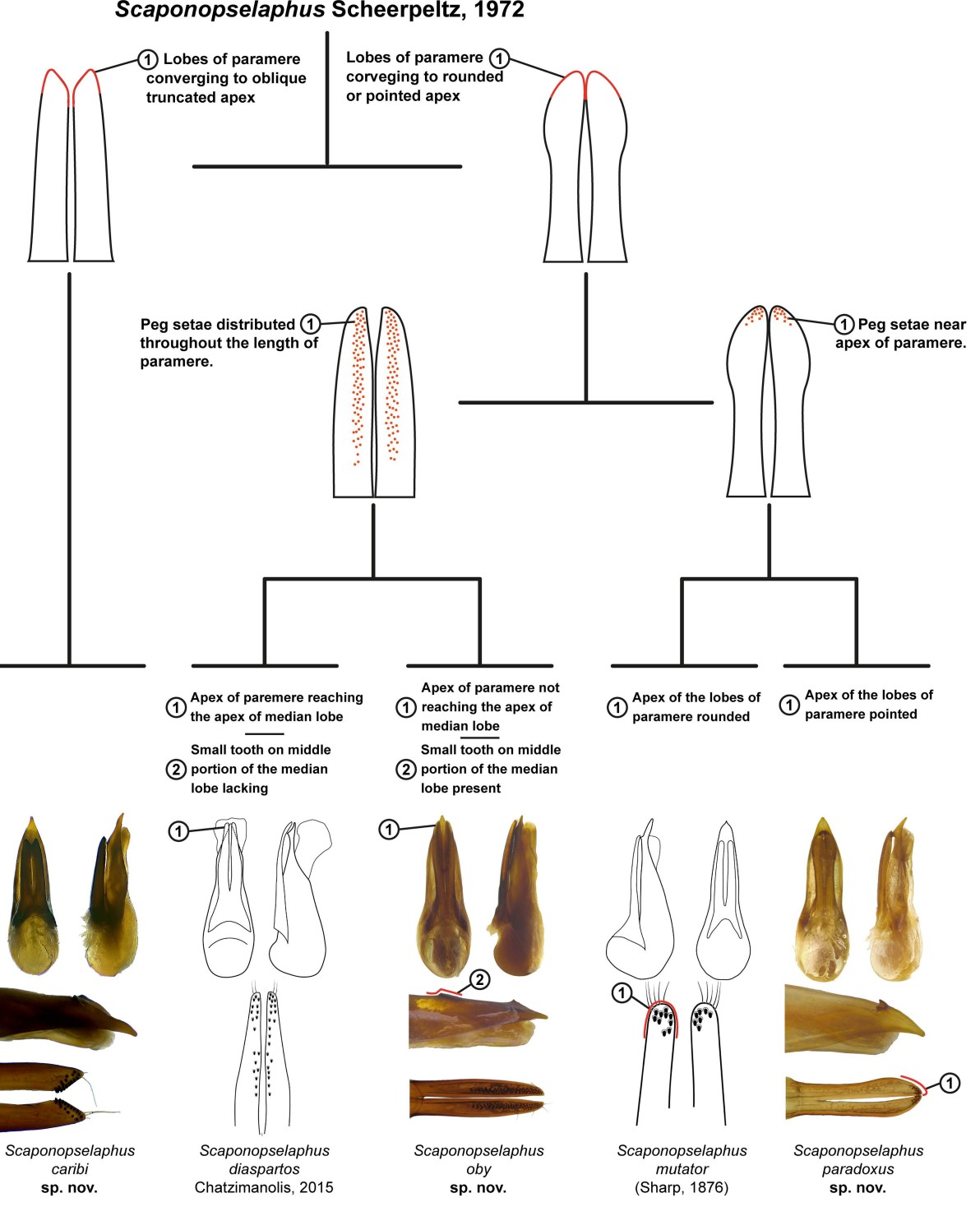

## *Scaponopselaphus* Scheerpeltz, 1972

① Lobes of paramere converging to oblique truncated apex

Lobes of paramere corveging to rounded or pointed apex ①

① Peg setae distributed throughout the length of paramere.

Peg setae near apex of paramere. ①

① Apex of paremere reaching the apex of median lobe

② Small tooth on middle portion of the median lobe lacking

① Apex of paramere not reaching the apex of median lobe

② Small tooth on middle portion of the median lobe present

① Apex of the lobes of paramere rounded

① Apex of the lobes of paramere pointed

*Scaponopselaphus caribi*
**sp. nov.**

*Scaponopselaphus diaspartos*
Chatzimanolis, 2015

*Scaponopselaphus oby*
**sp. nov.**

*Scaponopselaphus mutator*
(Sharp, 1876)

*Scaponopselaphus paradoxus*
**sp. nov.**

**Fig 12. Pictorial Key to species of *Scaponopselaphus* Scheerpeltz.**

Antennomeres (Fig 9C) 1–3 longer than wide, gradually club-like thickened, and without microtrichae; antennomere 1 (AL: 0.5; AW: 0.2), antennomere 2 (AL: 0.29; AW: 0.15), antennomere 3 (AL: 0.26; AW: 0.16); antennomeres 4–10 transverse and just slightly asymmetrical, antennomere 4 (AL: 0.15; AW: 0.19), antennomere 5 (AL: 0.16; AW: 0.22), antennomere 6 (AL: 0.15; AW: 0.22), antennomere 7 (AL: 0.16; AW: 0.23), antennomere 8 (AL: 0.15; AW: 0.23), antennomere 9 (AL: 0.16; AW: 0.23), antennomere 10 (AL: 0.17; AW: 0.22); antennomere 11 longer than wide (AL: 0.29; AW: 0.2); antennomeres 4–11 densely covered by microtrichae. Dorsal surface of neck with microsculpture like vertex of head and with few micropunctures (Fig 9B).

**Pronotum** (Fig 9A–9B) wider than long (PL: 1.83; PW: 1.94), with anterior margin almost straight, lateral margin concave and hind lateral angles and posterior margin rounded. Disk with rounded umbilicate punctures, except on medial line that is abbreviated anteriorly; umbilicate punctures with light brown setae. Integument shining, with dense microsculpture in form of rooves between umbilicate punctures.

**Scutellum** (Fig 8A) with dense dark brown setae and micropunctures covering its surface.

**Elytra** (Fig 9A) wider than pronotum (EL: 2.24; EW: 2.52); surface polished and shiny; denser punctuate than pronotum; punctures almost confluent; covered with dark brown setae and, along borders with few dark brown macrosetae.

**Leg** (Fig 9A) with brown setae; protarsus dorsoventrally flattened and with spatulate pale setae ventrally; mesotarsomere 1 with spatulate pale setae ventrally (Fig 10F–10G). Mesotibia and metatibia with multiple rows of spurs on external side, and with apical spurs. Protibia without multiple rows of spurs on external side, and with single row of spurs apically.

**Abdomen** (Fig 9A) abdominal tergites III–VI with tergal basal carina; tergites III–V with subbasal curved carina. Sternite VII with round porose structure anteriorly (Fig 10A); its posterior margin with shallow and broad emargination at midline (Fig 10A); posterior margin of sternite VIII with deep U-shaped emargination at midline (Fig 10B); sternite IX with profound U-shaped emargination at midline (Fig 10C). Tergite IX long and straight covered with long dark brown setae and some dark macrosetae.

**Aedeagus** as in Fig 11A–11F (length 2.14 mm). Paramere (Fig 11B and 11F) divided longitudinally in two lobes narrower and shorter than median lobe (Fig 11B), and converging to pointed apex (Fig 11E–11F), which is slightly convex (Fig 11A) in lateral view, and with peg setae concentrated on apex of lobes, underside (Fig 11E–11F). Sides of median lobe converging to narrow pointed apex (Fig 11B and 11C); apical region with one small tooth on each connected by one carina, resembling a semicircle (Fig 11C–11D); median lobe tapering apically (Fig 11B–11C).

**Female**. Similar to male, but vertex of head slightly convex, posterior margin of sternites VII and VIII straight, and mesotarsomere 1 without spatulate pale setae ventrally.

**Habitat**. One specimen was collected with yellow trap in *Guadua*-mixed forest, and one specimen was collected with pitfall trap without bait in low hill forest respectively from Cuzco. There are no additional data for the others specimens.

**Known distribution** (Fig 2). Known from localities of Río Pisqui, Comunidad nativa Roaboya in the department of Loreto, and localities of Megantoni, San Martin Norte Sagari and Reserva Comunal Matsiguenga in the department of Cuzco, Peru.

**Etymology**. The specific name "*paradoxus*" is a noun in apposition.

**Remarks**. Between the male and female specimens studied, there is a variation in the metallic overtones, ranging from green to blue and purple-blue in the head and pronotum.

## Additional material examined

### *Scaponopselaphus mutator* (Sharp, 1876)

**French Guiana:** Itoupé, DZE 570, 03°01'23"N, 53°05'44"W, 800m, 17.iii.2010, *leg.* SEAG (1 male, UEA).

**Note:** Itoupé is a new locality record for French Guiana (Fig 2). The species was previously known from Brazil, Peru, Suriname and French Guiana [2]

**Key to species of *Scaponopselaphus* Scheerpeltz**

Key to species is provided in DELTA version in S1 File and pictorical key (Fig 12). Letter "C" before figure number refer to Chatzimanolis [2].

1. Apex of lobes of paramere oblique truncated, in parameral view (Fig 5E) … ***Scaponopselaphus caribi* n. sp.**

    - Apex of lobes of paramere rounded (Figs C6c and C7c) or pointed (Figs 8E and 11E), in parameral view … **2**

2. Peg setae scattered throughout the length of the lobes of paramere of aedeagus, in underside view (Fig 8E–8F) … **3**

    - Peg setae concentrated near the apex of the lobes of paramere of aedeagus, in underside view (Fig 11E–11F) … **4**

3. Apex of paramere of aedeagus, in lateral view, not reaching the apex of median lobe (Fig 8A–8B); one small tooth on the middle portion of the median lobe (Fig 8C–8D) … ***Scaponopselaphus oby* n. sp.**

    - Apex of paramere of aedeagus, in lateral view, reaching the apex of median lobe (Fig 6A–6B in Chatzimanolis [2]); small tooth on the middle portion of the median lobe lacking … ***Scaponopselaphus diaspartos* Chatzimanolis**

4. Apex of the lobes of paramere of aedeagus rounded (Fig 7B–7C in Chatzimanolis [2]) … ***Scaponopselaphus mutator* (Sharp)**

    - Apex of the lobes of paramere of aedeagus pointed (Figs 11E–11F) … ***Scaponopselaphus paradoxus* n. sp.**

**Biological notes**

The species of *Scaponopselaphus* were collected in evergreen Tropical Forests, *Guadua*-mixed forests, with Winkler, Pitfall trap, Moerick trap, Pink pan trap or flight intercept traps. Until now, nothing is known about his food preferences.

## Discussion

### Taxonomy

The new species belong to the genus *Scaponopselaphus*, based on the presence of the following characters used by Chatzimanolis [2]: head with distinctive microsculpture (Figs 3B,6B and 9B); labial palpomere 3 securiform (Figs 4D–4E, 7D–7F and 10D–10E); pronotum with broad and convex lateral margins (Figs 3A–3B,6A–6B and 9A–9B); mesotarsomere 1 in males with spatulate setae (Figs 4F–4G,7G–7H and 10F–10G); tergites III–V with curved carina (Figs 3A,6A and 9A), and sternite VII in males with porose structure (Figs 4A,7A and 10A).

### Distribution and biological notes

Currently, biological information on *Scaponopselaphus* Scheerpeltz is limited, likely due to factors such as observational challenges, a lack of focused studies on the genus, and the rarity of its specimens in scientific collections. This lack of knowledge hinders the ability to conduct targeted sampling, as there are no records of the specific environments occupied. Some inferences, however, can be drawn from the genus morphological traits. The morphology among species is highly conserved, with differences observed almost exclusively in male genitalia, which suggests that species within the genus occupy similar ecological niches. Other inferences can be drawn from the field sample data, since the majority of the species of *Scaponopselaphus* collected have been obtained from intercepted flights traps of tropical forests, which highlights the effectiveness in sampling these species with these methods, however, there is still doubts about the habitat occupied since flight interception traps may collect taxa from a variety of environments such as canopy and understory. Further targeted efforts exploring target regions could help knowledge gaps in taxonomy, distribution and biology, providing a better understanding of the genus and its ecological requirements. Such research will be crucial for guiding future conservation and ecological studies.

## Supporting information

**S1 Table. Measurements of specimens of *Scaponopselaphus caribi* n. sp., *Scaponopselaphus oby* n. sp. and *Scaponopselaphus paradoxus* n. sp., comparing the holotype specimen with the paratypes.**
(XLSX)

**S1 File. Key to species of the genus *Scaponopselaphus* Scheerpeltz in DELTA (DEscription Language for TAxonomy) version .**
(DLT)

**S2 File. Distribution map of the species of *Scaponopselaphus* Scheerpeltz in the KMZ version.**
(KML)

## Acknowledgments

We would like to thank Conselho Nacional de Desenvolvimento Científico e Tecnológico (CNPQ) and Coordenação de Aperfeiçoamento de Pessoal de Nível Superior (CAPES) for the constant financial support granted to LKJ, and Precious Woods – Mil Madeiras Preciosas for the logistical support that made it possible to capture specimens of the species described here. We appreciate the Laboratório de Invertebrados (LA-INV) assistance and support of our colleagues, with special acknowledgment to Bruna Façanha for her support and patience throughout the process. The participation of AA was funded by CONCYTEC through the PROCIENCIA program within the framework of the call "Interinstitutional Alliances for Doctorate Programs" according to contract PE501084299-2023-PROCIENCIA-BM. The authors are grateful to the Norsk-Hydro and the Biodiversity Research Consortium (BRC) for their financial support provided through the projects "Diversity of the herbivorous insects in four areas of the Hydro mining company, Paragominas, Pará, Brazil" and "Metabarcoding and metagenomics for high-throughput inventory and monitoring of terrestrial arthropod diversity." This paper is number BRC0073 in the publication series of the Biodiversity Research Consortium Brazil-Norway (BRC).

## Author contributions

**Conceptualization:** Louri Klemann-Junior, Angélico Asenjo, Bruno Gouvea, Roberta M. Valente.

**Data curation:** Louri Klemann-Junior, Angélico Asenjo, Bruno Gouvea, Roberta M. Valente.

**Formal analysis:** Louri Klemann-Junior, Angélico Asenjo, Bruno Gouvea, Roberta M. Valente.

**Investigation:** Louri Klemann-Junior, Angélico Asenjo, Bruno Gouvea, Roberta M. Valente.

**Methodology:** Louri Klemann-Junior, Angélico Asenjo, Bruno Gouvea, Roberta M. Valente.

**Supervision:** Louri Klemann-Junior, Angélico Asenjo, Bruno Gouvea, Roberta M. Valente.

**Validation:** Louri Klemann-Junior, Angélico Asenjo, Bruno Gouvea, Roberta M. Valente.

**Writing – original draft:** Louri Klemann-Junior, Angélico Asenjo, Bruno Gouvea, Roberta M. Valente.

**Writing – review & editing:** Louri Klemann-Junior, Angélico Asenjo, Bruno Gouvea, Roberta M. Valente.

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
