## [Decision Letter · Decision Letter 0]

Dear Dr. Klemann-Junior,

Thank you for submitting your manuscript to PLOS ONE. After careful consideration, we feel that it has merit but does not fully meet PLOS ONE’s publication criteria as it currently stands. Therefore, we invite you to submit a revised version of the manuscript that addresses the points raised during the review process.

We look forward to receiving your revised manuscript.

Kind regards,

Alex Slavenko

Academic Editor

PLOS ONE

2. We note that Figures 2, 13 and 14 in your submission contain [map/satellite] images which may be copyrighted. All PLOS content is published under the Creative Commons Attribution License (CC BY 4.0), which means that the manuscript, images, and Supporting Information files will be freely available online, and any third party is permitted to access, download, copy, distribute, and use these materials in any way, even commercially, with proper attribution. For these reasons, we cannot publish previously copyrighted maps or satellite images created using proprietary data, such as Google software (Google Maps, Street View, and Earth). For more information, see our copyright guidelines: http://journals.plos.org/plosone/s/licenses-and-copyright.

1. You may seek permission from the original copyright holder of Figures 2, 13 and 14 to publish the content specifically under the CC BY 4.0 license. 

Additional Editor Comments:

I have now received three independent reviews of this manuscript. Reviewers #1 and #2 both have high praise for the taxonomic sections of the manuscript, complementing the thoroughness of the work undertaken. Reviewers #1 and #3 have some suggestions to improve the clarity and correct some errors, and reviewer #3 especially has submitted some very thorough edits for the manuscript. I encourage the authors to go over these suggestions carefully prior to resubmission.

I do share reviewer #2's concerns regarding the ENMs, particularly in regards to some of the methodological decisions made and how these might bias the results and interpretation. However, I also agree with the reviewer that the entire section could be removed from the manuscript to no great detriment, and it would still be a quality manuscript worthy of publication. Ultimately, I leave the decision up to the authors on whether to attempt to revise the ENM section according to reviewer 2's suggestions or delete it entirely. Either would be acceptable.

Reviewers' comments:

Reviewer's Responses to Questions

**Comments to the Author**

1. Is the manuscript technically sound, and do the data support the conclusions?

Reviewer #1: Yes

Reviewer #2: Partly

Reviewer #3: Yes

2. Has the statistical analysis been performed appropriately and rigorously?

Reviewer #1: Yes

Reviewer #2: No

Reviewer #3: I Don't Know

3. Have the authors made all data underlying the findings in their manuscript fully available?

Reviewer #1: Yes

Reviewer #2: Yes

Reviewer #3: Yes

4. Is the manuscript presented in an intelligible fashion and written in standard English?

Reviewer #1: Yes

Reviewer #2: Yes

Reviewer #3: Yes

Reviewer #1: This is a great taxonomic paper describing three distinct new species of Scaponopselaphus. The illustrations for various species are really well done. There are no major issues with this paper and I think it is definitely worthy of publication.

I have several minor comments that I think can improve the readability and usefulness of this paper.

1. There are a lot of measurement data in this paper, especially data that are usually absent in rove beetle papers (e.g. individual antennomere measurements). There is frequently a lot of developmental variability and I am not convinced of the usefulness of these data. These measurements are already provided as supplemental materials and I would simply remove them from the main paper, because frankly, nobody is going to read or use these.

2. I would like to caution the authors that sometimes it is very hard to tell the difference between black and dark brown color and several factors can influence this (variation in the coloration of a specimen, preservation methods, how long a specimen was in fluid before pinned, how old is the specimen). I do not think the difference between black ad dark brown is enough to differentiate the species in the key, especially between new species 1 and 2 presented here. I understand that there very few other morphological characters other than the aedeagus.

3. The GBIF occurrence data is not independent from the data in the Chatzimanolis revision since 19/39 are from that revision (data from GBIF). This is not entirely clear in the Material and Methods section. Given the limited number of independent occurrences (25) used for the ecological niche modeling analysis, I would recommend downplaying the importance of the analyses here, especially given that we have no idea of the the specific habitat that these beetles prefer. While I appreciate the attempt to bring something different to a taxonomic paper, the reality is that we did not get any new insights from these analyses that one cannot get simply by glancing at the distribution map.

4. A few specimens are labeled as sp. in the map. What are those? Are those potential new species or specimens that cannot be identified? Perhaps they should be mentioned somewhere in the text.

Reviewer #2: My recommendation is for major revisions, mainly due to the issues in the ecological niche modeling (ENM) section. That said, I want to congratulate the authors on an outstanding taxonomic revision. The inclusion of pictographic keys and the excellent photographic documentation make this a valuable contribution to the knowledge of such a fascinating group. This work absolutely deserves publication, even without the ENM section. As I explain in more detail in my comments within the manuscript, if the data do not support species-level niche models and the suggested methodological improvements are not considered, I recommend removing this section to maintain the overall strength of the study.

Reviewer #3: The manuscript has several errors to correct

- the numbering of some figures (especially Figs 4 and 5 exchanged in the text)

- the measurements and averages. If the authors have used a method that takes into account the number of specimens examined to obtain the average values, they must explain it in the chapter Measurement or the values must be corrected as in the attached file

-It is not clear which character of the aedeagus distinguishes Scaponopselaphus paradoxus. It would be good to indicate it with an arrow in the figure.

I am not an English language specialist, but I think a revision in this sense would be appropriate

**Do you want your identity to be public for this peer review?** For information about this choice, including consent withdrawal, please see our Privacy Policy

Reviewer #1: **Yes: ** Stelios Chatzimanolis

Reviewer #2: No

Reviewer #3: **Yes: ** Giorgio Sabella

---

## [Author Response · Author response to Decision Letter 1]

7 May 2025

Thanks for the valuable comments. They were almost entirely incorporated into the manuscript.

Following are a point by point response (in green) to each of the points made by the Reviewers.

Response: The manuscript has been reviewed for style requirements.

2. We note that Figures 2, 13 and 14 in your submission contain [map/satellite] images which may be copyrighted. All PLOS content is published under the Creative Commons Attribution License (CC BY 4.0), which means that the manuscript, images, and Supporting Information files will be freely available online, and any third party is permitted to access, download, copy, distribute, and use these materials in any way, even commercially, with proper attribution. For these reasons, we cannot publish previously copyrighted maps or satellite images created using proprietary data, such as Google software (Google Maps, Street View, and Earth). For more information, see our copyright guidelines: http://journals.plos.org/plosone/s/licenses-and-copyright.

Response: Figures 13 and 14 have been removed from the manuscript. Images used to produce the distribution map (Fig. 2) were obtained from public domain datasets: Natural Earth, for terrain elevation and country boundaries; and the Brazilian Institute of Geography and Statistics (IBGE), for Brazilian states boundaries.

Additional Editor Comments:

I have now received three independent reviews of this manuscript. Reviewers #1 and #2 both have high praise for the taxonomic sections of the manuscript, complementing the thoroughness of the work undertaken. Reviewers #1 and #3 have some suggestions to improve the clarity and correct some errors, and reviewer #3 especially has submitted some very thorough edits for the manuscript. I encourage the authors to go over these suggestions carefully prior to resubmission.

I do share reviewer #2's concerns regarding the ENMs, particularly in regards to some of the methodological decisions made and how these might bias the results and interpretation. However, I also agree with the reviewer that the entire section could be removed from the manuscript to no great detriment, and it would still be a quality manuscript worthy of publication. Ultimately, I leave the decision up to the authors on whether to attempt to revise the ENM section according to reviewer 2's suggestions or delete it entirely. Either would be acceptable.

Response: Thanks for the valuable comments. They were almost entirely incorporated into the manuscript. Following are a point by point response (in green) to each of the points made by the Reviewers.

Reviewers' comments:

Review Comments to the Author

Reviewer #1: This is a great taxonomic paper describing three distinct new species of Scaponopselaphus. The illustrations for various species are really well done. There are no major issues with this paper and I think it is definitely worthy of publication.

I have several minor comments that I think can improve the readability and usefulness of this paper.

1. There are a lot of measurement data in this paper, especially data that are usually absent in rove beetle papers (e.g. individual antennomere measurements). There is frequently a lot of developmental variability and I am not convinced of the usefulness of these data. These measurements are already provided as supplemental materials and I would simply remove them from the main paper, because frankly, nobody is going to read or use these.

Response: We excluded from the manuscript measurements of structures not normally measured. We retained these measurements in the supplementary materials.

2. I would like to caution the authors that sometimes it is very hard to tell the difference between black and dark brown color and several factors can influence this (variation in the coloration of a specimen, preservation methods, how long a specimen was in fluid before pinned, how old is the specimen). I do not think the difference between black ad dark brown is enough to differentiate the species in the key, especially between new species 1 and 2 presented here. I understand that there very few other morphological characters other than the aedeagus.

Response: We excluded color from species differentiation and focused on aedeagus characteristics.

3. The GBIF occurrence data is not independent from the data in the Chatzimanolis revision since 19/39 are from that revision (data from GBIF). This is not entirely clear in the Material and Methods section. Given the limited number of independent occurrences (25) used for the ecological niche modeling analysis, I would recommend downplaying the importance of the analyses here, especially given that we have no idea of the specific habitat that these beetles prefer. While I appreciate the attempt to bring something different to a taxonomic paper, the reality is that we did not get any new insights from these analyses that one cannot get simply by glancing at the distribution map.

Response: We removed the ecological niche modeling analysis from the manuscript.

4. A few specimens are labeled as sp. in the map. What are those? Are those potential new species or specimens that cannot be identified? Perhaps they should be mentioned somewhere in the text.

Response: Records of Scaponopselaphus sp. refer to female specimens, whose species identification was not possible using only external morphological characters. We included this text in the manuscript.

Reviewer #2: My recommendation is for major revisions, mainly due to the issues in the ecological niche modeling (ENM) section. That said, I want to congratulate the authors on an outstanding taxonomic revision. The inclusion of pictographic keys and the excellent photographic documentation make this a valuable contribution to the knowledge of such a fascinating group. This work absolutely deserves publication, even without the ENM section. As I explain in more detail in my comments within the manuscript, if the data do not support species-level niche models and the suggested methodological improvements are not considered, I recommend removing this section to maintain the overall strength of the study.

Response: We removed the ecological niche modeling analysis from the manuscript.

Reviewer #3: The manuscript has several errors to correct

- the numbering of some figures (especially Figs 4 and 5 exchanged in the text)

Response: We checked the numbering of figures throughout the manuscript and corrected any errors found.

- the measurements and averages. If the authors have used a method that takes into account the number of specimens examined to obtain the average values, they must explain it in the chapter Measurement or the values must be corrected as in the attached file

Response: To improve clarity, we have excluded the averages and kept only the maximum and minimum measurements found.

-It is not clear which character of the aedeagus distinguishes Scaponopselaphus paradoxus. It would be good to indicate it with an arrow in the figure.

Response: The indication of the difference in the aedeagus of S. paradoxus is in figure 12.

I am not an English language specialist, but I think a revision in this sense would be appropriate

Response: The manuscript has undergone a new English language revision.

---

## [Decision Letter · Decision Letter 1]

An update on Scaponopselaphus Scheerpeltz (Coleoptera: Staphylinidae) with the description of three new species, and a key to the species

PONE-D-25-05254R1

Dear Dr. Klemann-Junior,

We’re pleased to inform you that your manuscript has been judged scientifically suitable for publication and will be formally accepted for publication once it meets all outstanding technical requirements.

Kind regards,

Alex Slavenko

Academic Editor

PLOS ONE

Additional Editor Comments (optional):

The manuscript has now been seen by one of the original reviewers, who confirmed that all of their comments have been adequately addressed. Based on the reviewer's opinion and my own reading of the manuscript, I agree that the manuscript can be accepted for publication.

Reviewers' comments:

Reviewer's Responses to Questions

**Comments to the Author**

Reviewer #2: All comments have been addressed

2. Is the manuscript technically sound, and do the data support the conclusions?

Reviewer #2: Yes

3. Has the statistical analysis been performed appropriately and rigorously?

Reviewer #2: N/A

4. Have the authors made all data underlying the findings in their manuscript fully available?

Reviewer #2: Yes

5. Is the manuscript presented in an intelligible fashion and written in standard English?

Reviewer #2: Yes

Reviewer #2: (No Response)

**Do you want your identity to be public for this peer review?** For information about this choice, including consent withdrawal, please see our Privacy Policy

Reviewer #2: No

---

## [Editor Report · Acceptance letter]

PONE-D-25-05254R1

PLOS ONE

Dear Dr. Klemann-Junior,

I'm pleased to inform you that your manuscript has been deemed suitable for publication in PLOS ONE. Congratulations! Your manuscript is now being handed over to our production team.

Kind regards,

on behalf of

Dr. Alex Slavenko

Academic Editor

PLOS ONE